# Knowledge-Augmented Large Language Models for Personalized Contextual Query Suggestion

## ABSTRACT

Large Language Models (LLMs) excel at tackling various natural language tasks. However, due to the significant costs involved in re-training or fine-tuning them, they remain largely static and difficult to personalize. Nevertheless, a variety of applications could benefit from generations that are tailored to users' preferences, goals, and knowledge. Among them is web search, where knowing what a user is trying to accomplish, what they care about, and what they know can lead to improved search experiences. In this work, we propose a novel and general approach that augments an LLM with relevant context from users' interaction histories with a search engine in order to personalize its outputs. Specifically, we construct an entity-centric knowledge store for each user based on their search and browsing activities on the web, which is then leveraged to provide contextually relevant LLM prompt augmentations. This knowledge store is light-weight, since it only produces user-specific aggregate projections of interests and knowledge onto public knowledge graphs, and leverages existing search log infrastructure, thereby mitigating the privacy, compliance, and scalability concerns associated with building deep user profiles for personalization. We then validate our approach on the task of contextual query suggestion, which requires understanding not only the user's current search context but also what they historically know and care about. Through a number of experiments based on human evaluation, we show that our approach is significantly better than several other LLM-powered baselines, generating query suggestions that are contextually more relevant, personalized, and useful.

## CCS CONCEPTS

• **Computing methodologies** → **Natural language processing**; • **Information systems** → *Personalization*.

## KEYWORDS

Large Language Models, Personalization, Entity-centric Knowledge, Contextual Query Suggestion

**ACM Reference Format:**

Anonymous Author(s). 2023. Knowledge-Augmented Large Language Models for Personalized Contextual Query Suggestion. In *Proceedings of Make sure to enter the correct conference title from your rights confirmation emai (Conference acronym 'XX).* ACM, New York, NY, USA, 13 pages. https://doi.org/XXXXXXX.XXXXXXX

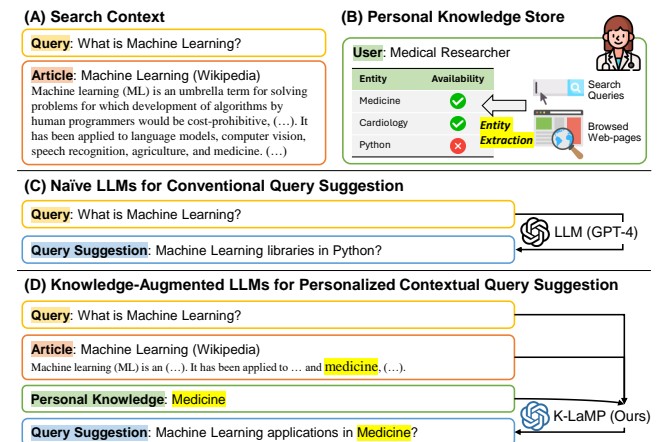

**Figure 1: Illustration of our proposed Knowledge-augmented large Language Model for Personalization (K-LaMP) framework for our contextual query suggestion task. (A) A user's search context includes a current query and an article being viewed. (B) The user has knowledge of medicine, extracted from past search activities. (C) The conventional query suggestion with naïve LLMs generates a query unrelated to the user's knowledge and the context. (D) Our K-LaMP suggests a pertinent query that is personally and contextually related.**

## 1 INTRODUCTION

Large Language Models (LLMs) [6, 9, 16, 53, 68, 69], such as GPT-4, are multi-billion parameter models trained on massive text corpora, which are capable of internalizing general knowledge across diverse domains [55, 59]. This capability allows them to generate plausible, reasonable, and helpful outputs in response to user inputs, and has been leveraged with impressive results for a diverse range of natural language tasks, including question answering and dialogue generation, even without any task-specific training [6, 53, 69].

Despite these successes, customizing LLMs to generate personalized responses, which take into account the individual preferences, needs, knowledge and context of users, with the goal of making them more meaningful and relevant to each user, remains challenging. This is due to the fact that re-training or fine-tuning LLMs for individual users is prohibitively expensive. On the other hand, for some real-world applications, such as query suggestion [22, 64], item recommendations [26, 76], snipped generation [13] or question answering [31], reflecting personal preferences, knowledge and needs of users in the model's outputs is essential.

Several recent studies [20, 29, 32, 36, 46, 48] have tackled the problem of LLM personalization through augmenting the user's input with relevant information, a process known as in-context learning [9, 39, 58, 74]. For example, in order to suggest the next item that the user may interact with, they prepend a sequence of their past item interactions into the LLM's input [14, 46]. Yet, due

to the often large volume of users' historical information, more recent work [60, 63, 72, 78] proposes to inject only a fraction of the most relevant history by retrieving it from a complete interaction memory. Such retrieval-based personalization, while demonstrating successes on several tasks including product recommendation and writing assistance [14, 44, 60, 72], remains ill-suited for more challenging personalization scenarios where a deeper understanding of users' personal *knowledge* is crucial. Meanwhile, some studies [38, 79] enable personalization through construction of deep user profiles that are then incorporated into LLM prompts. Nevertheless, such profile-based personalization captures user knowledge at the cost of privacy and scalability, requiring online modeling beyond the capabilities of existing logging infrastructure.

Thus, in this paper, we introduce a novel approach to personalizing the output of LLMs. Our method revolves around *an entity-centric light-weight personalization layer* that enables knowledge-augmentation of LLMs with contextual entities retrieved from a *personal knowledge store*. This knowledge store is derived from existing search logs that capture users' interactions with modern search engines. Specifically, this store is built over time by aggregating entities that appeared in queries that the user issued, or web-pages that they browsed, and is further enhanced by different views that capture the entities the user may be familiar or unfamiliar with, and those that may have recently lapsed from their memory.

This entity-centric output personalization strategy has several advantages. First, contextually relevant retrievals from this entity-centric knowledge store encourage LLMs to generate outputs that are more deeply grounded in what users know and care about as compared with linearly stored past query logs. At the same time, it largely relies on already existing logging infrastructure, which means that it is more amenable to privacy, flexibility and scalability considerations than profile-based personalization, as it reduces data collection, modeling and update overheads. Also, thanks to its light-weight design, our approach offers easy integration with existing LLMs for other personalization tasks, such as in search [13, 31] and beyond [44, 60]). Finally, our knowledge augmentation method is cost efficient since the knowledge injection employs entities as atoms. This results in minimal, succinct additions to the prompt, unlike other LLM contextualization approaches that operate over raw texts [39, 58, 74]. We refer to our framework as **K**nowledge-augmented large **La**nguage **M**odels for **P**ersonalization, or K-LaMP.

While our method of personalizing LLM outputs is broadly applicable to problems in search (and beyond), it is especially relevant to tasks that require modeling the *knowledge* of users in addition to their interests. One such challenging task is a new variant of query suggestion [4, 12, 23, 65], that we call *contextual* query suggestion. In this variant, a system must recommend queries to a user conditioned on a web-page they are currently reading, in addition to their historical query information. Thus, in this setting, knowing a user's domain of expertise and proficiency about a particular topic can lead to substantially different suggestions, as shown in Figure 1. It is worth noting that contextual query suggestion is different from existing *context-aware* query suggestion in the literature [5, 15, 34, 37], since the latter neither conditions recommendations on the body of the web-page being viewed by the user, nor explicitly captures the user's knowledge, instead focusing on surface-level relationships between queries and pages, or their titles.

In our investigation, we validate the effectiveness of K-LaMP for contextual query suggestion, using real-world search logs from the public search engine [2], which is also used to construct our entity-centric knowledge store. On a battery of tests conducted via human evaluation, we find that K-LaMP substantially outperforms several LLM-powered (contextual) query suggestion baselines in generating recommendations that are better related and more useful to individuals, while maintaining high search result quality. Further analyses demonstrate that K-LaMP retrieves contextually relevant knowledge in a highly effective manner, and continues to become more performant as longer user interaction histories are processed and stored, neither of which other baselines are capable of doing.

## 2 BACKGROUND AND RELATED WORK

**Large Language Models.** Language models [24, 47, 56, 57], which are pre-trained on unannotated text corpora with Transformer architectures [70] based on self-supervised learning objectives, have been shown to acquire knowledge from text corpora [55, 59, 66] and successfully used for various natural language tasks, such as question answering and dialogue generation [35, 67]. Recently, Large Language Models (LLMs) [6, 53, 69], which are scaled-up version of language models, have demonstrated the capability of handling diverse language tasks across various domains. In particular, LLMs have shown increased capacity for knowledge acquisition and retention thanks to their very large number of parameters [49, 77], as well as a remarkable ability to generalize across new domains with no need for additional task-specific fine-tuning and training data [61, 73]. Moreover, they are able to understand the context of given inputs and then generate contextually coherent responses, allowing users and system designers to easily customize LLMs through prompt engineering [39, 58, 74]. For example, to generate factually correct answers in response to input questions, existing work [7, 40, 62] typically augments the internalized knowledge in LLMs with externally relevant factual knowledge related to questions. However, while they can effectively provide *generic responses* that may apply to a broad swath of users, it remains challenging to generate *personalized responses* that capture the unique preferences, needs, and knowledge of individual users.

**Large Language Models for Personalization.** In order to yield outputs that are customized to individual users, recent studies [10, 45] propose to personalize the generations of LLMs, with applications spanning various tasks and domains. These include product or content recommendations [14, 27, 30, 72], dialogue generations [63, 78], writing assistants [44, 60], and even robotic systems [75]. Specifically, early work [20, 29, 32, 36, 46, 48] proposes to incorporate the historical sequence of the user's interactions (e.g., recent purchase logs of items) into LLMs prompts, thereby allowing LLMs to generate outputs that are personalized (e.g., next item recommendation). While this simple, linear injection mechanism can effectively provide LLMs with relevant contextual information for personalization, it is limited by often very large interaction histories which exceed the capacity of LLM prompt windows. Also, not all of this history is relevant to every query. Based on this observation, recent work [60, 63, 72, 78] proposes to retrieve relevant content from an external memory [54] that stores the user's historical information. A few studies [44, 78] go a step further, processing the

information in the interaction memory – for example, with summarization or key-word extraction – to gain higher-level insights from the user's history when contextualizing LLMs for personalization.

Unlike prior work in LLM output personalization that focuses largely on modeling the *interests* of users, our work additionally targets their *knowledge* over topics and domains of interest. Accordingly, rather than only leveraging the linearly stored interaction histories of users, in this work, we build a personal knowledge store consisting of entities mined from search queries and page visitations. This mechanism, which provides a lens through which user knowledge can be captured, has two additional advantages: it enables light-weight personalization by retrieval from the knowledge store without requiring explicit profiling of users [38, 79]; the knowledge represented as entities is succinct, thereby leading to efficiency gains through reductions in input context length when compared with existing LLM contextualization work [58].

**Search Query Suggestion.** The goal of query suggestion is to recommend new queries of potential interest to users, based on current and previous queries in and across search sessions. This task is both highly practical and useful, having been shipped in web-scale search engines (Google and Bing), as well as been widely applied to other tasks and domains, such as task-oriented search [25, 28] and recruitment platforms [81]. Early work on query suggestion has used frequency-based statistical (probabilistic) methods, which include Markov or LDA models [11, 33, 37, 50, 71]. More recently, neural network methods based on recurrent or attention-based architectures [12, 23, 34, 65, 80] have been leveraged to better model past query sequences and generalize to unseen and long-tail queries. Meanwhile, other studies have proposed to improve training strategies by performing either multi-task learning with a document ranker [4, 5, 15] or reinforcement learning [8]. Finally, other recent work [51, 52] uses pre-trained language models [24, 43] to achieve superior performances with larger model capacity.

In comparison to prior work on query suggestion, we tackle the novel but realistic task of *contextual* query suggestion, where recommendations are additionally conditioned on the web-page a user is currently viewing. This task is clearly different from existing query suggestion work that leverages previously clicked pages [5, 15, 34, 37] only through surface-level association (such as relationships between past queries and page titles), because it requires contextualizing the full text of the page. This novel task is of particular interest to us because it exposes the need for personalized models that recommend queries based not only on what users are interested in, but also on what and how much they *know*.

## 3 K-LAMP: KNOWLEDGE-AUGMENTED LLMS FOR PERSONALIZED QUERY SUGGESTION

In this section, we introduce our approach to generating personalized outputs using a novel knowledge-augmentation method for LLMs and our entity-centric personal knowledge store for it, and detail its application to a novel task of contextual query suggestion.

### 3.1 Problem Statement

We begin with preliminaries, formally introducing LLMs and the problem of contextual query suggestion.

**Large Language Models.** Let us define an LLM as a model, parameterized by a set of parameters $\theta$, that takes an input sequence of tokens $x = [x_1, x_2, ..., x_n]$ and a supplemental sequence of context tokens $c = [c_1, c_2, ..., c_k]$ as a prompt, and then generates an output sequence of tokens $y = [y_1, y_2, ..., y_m]$. Then, formally, the inference of an LLM can be summarized as: $y = \text{LLM}_\theta(x, c)$. Here, $\theta$ is typically pre-trained auto-regressively on massive text corpora and remains fixed; $x$ is a task dependant user issued prompt or set of instructions; and $c$ is some additional context provided by an auxiliary system that helps augment, ground, or otherwise improve the quality of the input, so that the LLM is able generate outputs $y$ more effectively. This paper particularly focuses on the nature of $c$ for the task of contextual query suggestion defined below, and with the use of entity-centric knowledge for more personalized outputs.

**Contextual Query Suggestion.** Before formalizing Contextual Query Suggestion that we introduce in this work, we first define a task of conventional Query Suggestion. Let $q_j$ be the most recent query issued by a user and $q_h = [q_1, q_2, ..., q_{j-1}]$ be a sequence of their historical queries. Then, a query suggestion model QS aims to predict new queries $q_{j+1}$ that an individual user with current query $q_j$ and query history $q_h$ might be likely to find useful. This process can be summarized as follows: $q_{j+1} = \text{QS}_\theta(q_j, q_h)$.

Contextual query suggestion expands on this definition to incorporate a broader set of context $c = [c_1, c_2, ..., c_k]$ linearized as sequences of text. Specifically, let us first assume that $x$ is an input query: $x = q_j$. Then, $q_h \in c$, meaning that the query history is one of the contextual signals capable of being leveraged for query suggestion. In this task, the text of a web-page $w$ currently being consumed by the user is also included in $c$, as follows: $w \in c$. Formally, this task can be summarized as follows: $q_{j+1} = \text{QS}_\theta(x, c)$[1]. 

In this work, we attempt to solve the problem of contextual query suggestion by leveraging a knowledge-augmented model to yield more personalized outputs. Formally, for $q_{j+1} = \text{QS}_\theta(x, c)$, we set $QS$ to be an LLM (e.g., GPT-4 [53]) and include aggregated entity-centric knowledge from users' historical interactions in the context $c$, in order to generate *better* recommendations $q_{j+1}$, as measured by a set of personalization-focused metrics (see Section 4.3).

### 3.2 Knowledge-Augmented LLMs for Personalization with Knowledge Store

We now discuss our knowledge-augmented LLM framework for personalization and describe two instantiations of this framework.

Recall that the supplemental context of an LLM $c$ is provided by auxiliary sources or systems that help enrich the input prompt to the model. According to our goal of personalizing the outputs of the LLM, these auxiliary sources should ideally consist of data that captures the personal preferences, interests, and knowledge of individual users. Thus, if $\mathcal{K}$ is a knowledge store that encapsulates these user-specific data, and $k \in \mathcal{K}$ is a contextually relevant subset linearized as text, then, for the task of contextual query suggestion, the context $c$ can be defined as follows: $c = [q_h \cdot w \cdot k]$, where $[\cdot]$ is the concatenation operation.

Given this general formulation, the important questions to answer are: (1) How is the personal knowledge store $\mathcal{K}$ constructed?

---

[1] Without the hard requirement of an input web document included in $c$, this definition may be relaxed to capture prior work on *context-aware* query suggestion [5, 15, 34, 37].

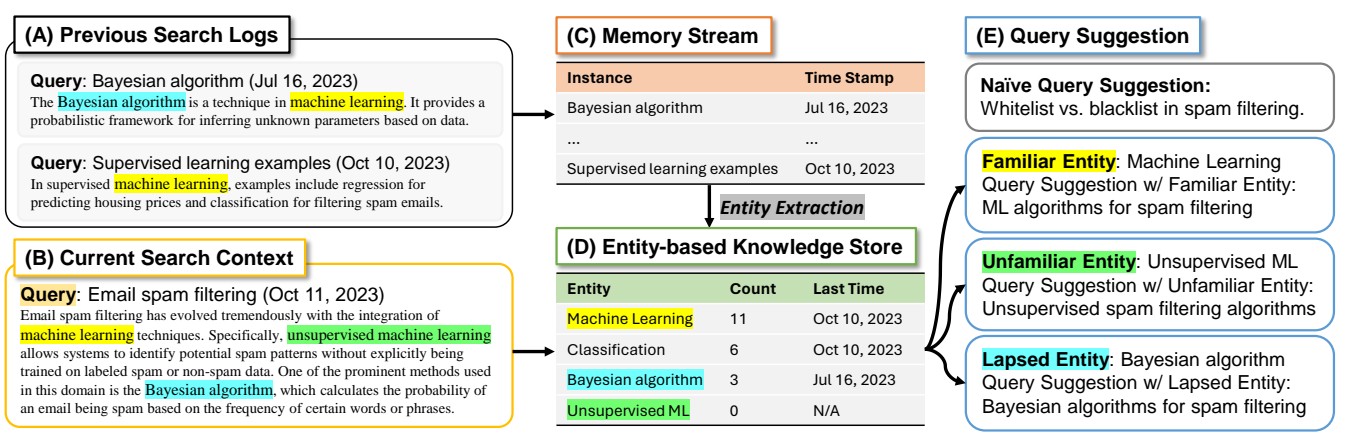

**Figure 2: Overview of K-LaMP. The inputs to the system are (A) previous search logs and (B) the current search context of a user, which consist of a query and an associated web-page. (C) Each search record is stored in a memory stream along with a time stamp. (D) The entity-based knowledge store is constructed by aggregating entities extracted from the memory stream. (E) K-LaMP augmented with entities retrieved from the personal knowledge store, generates query suggestions that are compatible with the user's knowledge and interests.**

and (2) How are contextually relevant items $k$ retrieved from the knowledge-store $\mathcal{K}$. We answer both of these questions below; namely, construction in Section 3.2.1 and retrieval in Section 3.2.2.

*3.2.1 Constructing Personal Knowledge Stores.* While a variety of sources may be used to construct a compendium of what a user cares about and knows, in this paper, we leverage users' interaction histories with a search engine. We argue that this is a natural choice given that users express goals, desires, interests, and depth of knowledge both explicitly and implicitly through issued queries, clicked web-pages, consumed content and behavioral patterns over time. It is also an especially relevant source of knowledge for contextual query suggestion, where generating helpful search suggestions from search histories is the eventual goal of the task. Given this source, we build two distinct instatiations of the knowledge store $\mathcal{K}$: a simple variant that linearly captures historical user queries and browsing patterns ($\mathcal{K}_s$), and another leveraging an entity-centric view of users' personal interests and knowledge ($\mathcal{K}_e$).

**Personal Knowledge Store from Search History ($\mathcal{K}_s$).** The intuition behind this instatiation of the knowledge store $\mathcal{K}_s$, is that users issue queries and click on web-pages that they are interested in or care about. And, when accumulated over time, these also start to construct a picture of what users know and how deeply do they know them. For example, a user that issues multiple queries over time including "Machine Learning", "ML", "Optimization", "SGD", "Deep Learning" and clicks a number of web-pages resulting from these queries, we can assume that they are at least familiar with the general concept of "Machine Learning".

In order to operationalize this intuition, we construct a time-stamped memory stream consisting of the queries issued by users and the web-pages associated with the results they clicked on (see Figure 2). Note that this is an extremely light-weight instatiation of the knowledge store, being only a partial view of user actions and interactions already logged by modern web-scale search engines. As a result, there are no privacy or scalability concerns beyond those already inherent in the search engine's logging system.

**Personal Entity-centric Knowledge Store ($\mathcal{K}_e$).** While building a memory stream over users' search histories is extremely simple, there are a few limitations that stem from its design. Firstly, because search queries and web-page visitations are stored and retrieved linearly, it is difficult to perform aggregations on the fly. Yet, such aggregation can be greatly beneficial for personalization. For example, knowing that a user clicked on web-pages associated with "Machine Learning" multiple times, while only clicking on a single web-page stemming from the query "Computational Biology" would tend to indicate a greater affinity for and knowledge of machine learning. Other issues include the fact that individual web-pages visited by a user may contain mixtures of several different topics and domains, distracting LLMs in generating outputs consistent to context, and the fact that including large amounts of text from lengthy web-pages renders LLM usage slow and expensive.

In order to address these concerns, we construct an entity-centric instance of the knowledge store $\mathcal{K}_e$ (see Figure 2). Entities are useful atoms for capturing the interests and knowledge of users because they consist of the nouns (proper or otherwise) that describe the people, places, organizations, topics and domains that the users care and know about. Additionally, because they tend to be relatively short and easy to aggregate, and because entity recognition and linking [18, 41] are well-studied problems, the process of operationalizing the creation of this store is greatly simplified.

Specifically, this is done by using a state-of-the-art entity linking system [19] to tag and canonicalize the entities that appear in the search queries and associated web-pages visited by users. While individual occurrences of entities in the knowledge store are time-stamped, additional aggregation is done by counting the number of occurrences of entities in entire user interaction histories.

Note that, while this entity-centric knowledge store instantiation $\mathcal{K}_e$ is not as minimalist as the instantiation $\mathcal{K}_s$, it is still relatively light-weight when compared with systems that personalize through the construction of deep profiles. The only external dependency is the entity linker, which can process thousands of tokens per second. In addition, scalability and privacy concerns are also small,

since entity linking projects onto sub-graphs of public entity graphs (e.g. Wikipedia), subsequently records can be easily removed upon request by eliminating associated entities from the store, and further aggregation of entity occurrences lends itself naturally to common privacy mitigation practises such as k-anonymization.

*3.2.2 Contextual Retrieval from Personal Knowledge Stores.* We now turn to the question of retrieving contextually relevant items $k$ from a knowledge-store $\mathcal{K}$, conditioned on the input query $q_j$ and web document $w$ that the user is currently interacting with. A carefully considered retrieval step is essential in augmenting the capability of LLMs to produce personalized outputs, since it grounds generation to historical interests and knowledge of users. In the following, we show how retrieval is performed for both instantiations of the knowledge store $\mathcal{K}_s$ and $\mathcal{K}_e$ described in Section 3.2.1.

In the case of $\mathcal{K}_s$ over users' search and browsing history, retrieval is done by finding and returning the most similar queries and previously visited web-pages to the current input $x$. In practise, the queries are then elided from this result since they yield little benefit over the much longer text present in web-pages. To operationalize the retrieval step, we first represent all records in the knowledge-store $\mathcal{K}_s$ using embeddings, then compute embedding-level similarities with the representation of current query $q_j$ using Contriever [42]. The most similar records $k$ are finally returned.

Meanwhile, for the entity-centric knowledge store $\mathcal{K}_e$, retrieval is conditioned on the entities present in the current query $q_j$ and the web-page $w$, which are further matched against $\mathcal{K}_e$. Given that entities are atomic units with associated counts and time-stamps, the matching and retrieval process can be operationalized in flexible ways (See Figure 2). We particularly explore three strategies for matching entities: familiar (entities the user has frequently encountered), unfamiliar (entities the user has encountered infrequently or not at all), and lapsed (entities that the user used to encounter previously but hasn't done so more recently). Specifically, for familiar entities, we sort the entities appearing in the search context $[x \cdot w]$ by frequency of occurrence in the knowledge store $\mathcal{K}_e$, then sample 5 entities proportionally to their frequency. For unfamiliar entities, a similar process is used for sampling, except that entities are sorted inversely with respect to their occurrence in $\mathcal{K}_e$. Finally, for lapsed entities, we start by filtering entities in $[x \cdot w]$ by time-stamp to retain only those that occur in $\mathcal{K}_e$, but haven't been engaged with in the preceding two weeks. Then we sample from this filtered set of entities by frequency, much like we do with familiar entities.

## 4 EXPERIMENTAL SETUP

In this section, we outline the datasets, models, evaluation setup, and implementation details.

### 4.1 Data

To validate K-LaMP for personalized contextual query suggestion task, we use real search logs from a large-scale web search engine [2]. Specifically, we sample three months of search logs, from May 01, 2023 to July 31, 2023. We then filter and sample this dataset to make it suitable for evaluating our task. First, because the task we are tackling is *contextual* query suggestion – i.e., recommendations are predicated on a current web-page the user is viewing – we filter out sessions that do not contain any clicked search results.

We further filter the data to discard click events that lead to pages in domains other than Wikipedia or a curated set of 500 high-traffic news publishers. We do this because the entity linker we use [19] maps onto Wikipedia, and we want to maximize the chances of encountering linked entities. It is worth noting that our K-LaMP framework itself is agnostic to the choice of the linker or its underlying knowledge graph, and our approach could readily be applied to a different domain, for exampling, using an entity linker over a product graph for shopping. Finally, we filter the remaining data to discard users who had fewer than 100 page visitations for three months, who we assume are infrequent users of the search engine. In addition, we perform and apply enterprise-level privacy checks and filters, such as using search queries requested from at least 50 individuals, to ensure that the data remains suitably anonymized.

The resulting data is still extremely large; therefore, we further randomly sample a subset of 1,000 users in order to get the benchmark set that forms the basis for all the evaluations we perform in this paper (Section 4.3). This final dataset contains, on average, 493 queries, 109 sessions, 177 clicked articles, and 3,053 encountered entities per user. For testing, we split the dataset and reserve the most recent 10 sessions of every user as prediction targets for contextual query suggestions and use all the earlier sessions for building search-and-browsing based ($\mathcal{K}_s$) and entity-centric ($\mathcal{K}_e$) personal knowledge stores for users, as described in Section 3.2.1.

### 4.2 Baselines and Our Model

We compare our approach to knowledge-augmented LLMs for output personalization against several relevant baselines that make query suggestions based on the search context of users. We note that, for comparisons, all baselines and our model use LLMs (specifically GPT-4) to make query suggestions. Also, prior query suggestion models based on traditional methods (e.g., RNNs or BART) [51, 65] are not directly applicable to the contextual query suggestion task due to their limited capacity for understanding longer context and complex data inputs, without dedicated training data.

The models evaluated in this work are listed as follows: (1) **Query Suggestion** – which uses a current query $q_j$ and historical queries from $q_h$ in the same session to suggest the next query $q_{j+1}$, (2) **Contextual Query Suggestion** – which is similar to Query Suggestion, but additionally conditions the recommendation of the next query $q_{j+1}$ on a web-page $w$, clicked as a result of current query $q_j$, (3) **Contextual Query Suggestion w/ $\mathcal{K}_s$** – which includes retrievals from the knowledge store $\mathcal{K}_s$ over users' historical search and browsing activities, as additional context to personalize the outputs of the LLM, and (4) **K-LaMP** – which is our full model that augments LLMs with entity-centric knowledge from the knowledge store $\mathcal{K}_e$ in order to perform contextual query suggestion.

### 4.3 Evaluation Setup

To evaluate the effectiveness of different query suggestion models on generating personalized outputs, a suitable evaluation metric should ideally not only capture whether the suggested queries are contextually relevant, but also whether they align well with the user's interests and knowledge. Given that contextual query suggestion is a novel problem we propose in this paper, there are no existing evaluation metrics for the task. In particular, metrics

**Table 1: Main results on our contextual query suggestion task. We emphasize the best results in bold.**

| Types | Models | Validness (↑) | Relatedness (↑) | Usefulness (↑) | Ranking (↓) |
|---|---|---|---|---|---|
| **Baselines** | Query Suggestion | 1.769 | 0.962 | 0.948 | 2.736 |
| | Contextual Query Suggestion | **1.966** | 1.267 | 1.245 | 2.415 |
| | Contextual Query Suggestion w/ $\mathcal{K}_s$ | 1.822 | 1.192 | 1.166 | 2.654 |
| **Ours** | **K-LaMP (Ours)** | **1.966** | **1.482** | **1.455** | **2.160** |

**Query**: Tim Cook
**Session**: Apple | Tim Cook
**Article**: Tim Cook Leadership
A new profile examines how Apple CEO Tim Cook, with "cautious, collaborative and tactical" leadership, honed the Cupertino tech giant into the world's largest company. (…)
**Trending entities**: 'GPT-4', 'OpenAI', 'Google Bard', 'Microsoft Copilot', 'Elon Musk', …
**Personal summary**: The user is interested in Apple products and technology ('Macbook', 'macOS', and 'Apple TV'). They have a keen interest in ML, with topics like 'Supervised Learning' and 'Optimization'. Additionally, they enjoy animation, showing interest in 'Studio Ghibli', 'Walt Disney', and 'Pixar'. Their preferences also extend to home entertainment ('DVD' and 'HDTV'). Lastly, they follow baseball ('MLB' and 'New York Yankees').
**Personal entities**: 'Macbook', 'macOS', 'Machine Learning', 'Optimization', 'Supervised Learning', 'Apple TV', 'Animation', 'Studio Ghibli', 'DVD', 'Walt Disney', 'Pixar Animation Studios', 'Apple Inc.', 'Baseball', 'HDTV', 'Major League Baseball', 'New York Yankees', …

**Figure 3: A fabricated example of the data that we provide to human judges.**

for evaluating conventional query suggestion are not applicable here because they do not account for the full *context* present in our task – namely the input document being consumed by the user. Therefore, we turn to human evaluation in order to measure and compare the different models on our experimental benchmark.

It is worth noting that human evaluation of any form of personalization is difficult, since the person performing the evaluation is rarely the person from whom the data originated. Short of flighting and A/B testing our system in a real-world setting – an engineering endeavor well beyond the scope of the scientific exploration in this paper – any evaluation on our task and dataset must be bounded by this constraint. Nevertheless, we attempt to provide annotators with as much information as possible in order to understand both the user's current search context and their personal interests and knowledge (see Figure 3 for the example). In particular, to summarize the current search context for annotators, we show them the current and previous search queries of a user in a given session, the web-page the user clicked on after issuing the current search query, and a list of 20 trending entities which capture statistical surges in search volume across users. Additionally, in order to present an encapsulation of the personal interests and knowledge of users, we show annotators a list of the 30 most frequent entities from the user's personal entity-centric knowledge store, as well as a GPT-4 generated summary from these entities that states what topics or domains the user may know or care about much.

Presented with these data and recommended queries from the different baselines and our model (where their names are obscured to annotators), a human judge is asked to evaluate the following three metrics on a 3-point Likert scale[2]: (1) **Validity** – whether an output query can be input into a search engine and be expected to yield relevant results; (2) **Relatedness** – whether the output query closely relates to the user's personal interests and knowledge; and (3) **Usefulness** – whether the user is likely to click on the output query, given their historical interests and knowledge as well as

[2]The 3-point Likert scale is composed of agree (2), neutral (1), and disagree (0).

**Table 2: Results of inter-annotator agreements on all query suggestion results evaluated by humans annotators.**

| Agreements | Metrics | Scores (↑) |
|---|---|---|
| Exact match | Validness | 0.963 |
| | Relatedness | 0.850 |
| | Usefulness | 0.819 |
| Cohen's kappa coefficient | Validness | 0.606 |
| | Relatedness | 0.652 |
| | Usefulness | 0.622 |
| Spearman's correlation coefficient | Ranking | 0.654 |

their current search context. Finally, we also ask the annotators for a fourth measure: (4) **Ranking** – where the outputs of the different systems are ranked according to the order in which they are likely to be clicked, based on their affinity to the user's interests, knowledge, and search context. Collectively, these four evaluation metrics capture not only how good the different query suggestions are, – both individually and in relation to one another – but also how well they align with the *personal* aspects of our evaluation task; namely, what users care about and know.

To perform evaluations with human judges, we recruit 12 annotators in India through a third-party vendor company [3]. They were provided with a guideline document, which includes instructions for the task, metrics and some annotated examples, and they were paid $11.98 per hour for the time they spent working on the task. Over several rounds of judgement and refinement, we obtain manual evaluation results for 1, 309 sets of contextual query suggestion results from all four models listed in Section 4.2 (effectively a total of 5, 236 annotations for individual query suggestions).

Additionally, in order to validate the quality of annotations and to measure inter-annotator agreement, approximately 27% of the data is annotated by two human judges. Specifically for Validity, Relatedness, and Usefulness, we measure an exact match score, which checks often annotators provide the same score on the 3-point likert scale, and Cohen's kappa coefficient [17] which additionally discounts for chance agreement. For Ranking, we report Spearman's correlation coefficient [1], which measures correlation between two sets of ranked systems, averaged across pairs of users and data instances. As shown in Table 2, we observe that inter-annotator agreement is moderate to high, indicating that judges are in fact able to make reasonably informed decisions about personalized contextual query suggestion from the data we provide them with.

### 4.4 Implementation Details

For a fair comparison, we use the GPT-4 [53] release from July 01, 2023, as the basis for query suggestion across all baselines and model

**Table 3: Results of different retrieval strategies on Retrieval Relevance.**

| Retrieval | Types | Retrieval Relevance ($\uparrow$) |
|---|---|---|
| History-based Retrieval ($\mathcal{K}_s$) | Past Documents | 0.299 |
| Entity-centric Retrieval ($\mathcal{K}_e$) | Familiar Entities | **0.936** |
| | Unfamiliar Entities | **0.810** |
| | Lapsed Entities | **0.849** |

variants. We set the hyperparameters of GPT-4 as temperature = 0.7 and top$_p$ = 0.95. The entity linker used to construct instantiations of the knowledge store is NEMO [18, 19][3]. Prompts used to elicit responses from GPT-4 for query suggestion are in Appendix A.

## 5 EXPERIMENTAL RESULTS

We now present the set of experimental results from our evaluation, and report findings from various auxiliary studies and analyses.

Our main results are shown in Table 1. This confirms that our K-LaMP framework consistently and significantly outperforms all other baselines across Relatedness, Usefulness, and Ranking metrics. While it ties Contextual Query Suggestion on the Validity metric, this finding is overall a positive and not an unexpected one – since intuitively, inclusion of personal context does not necessarily lead to queries that are more *valid* for search engine retrieval. Meanwhile, there are a few other interesting insights that can be gleaned from this table. Interestingly, Contextual Query Suggestion with $\mathcal{K}_s$ does not outperform Contextual Query Suggestion. We hypothesize that this is because the information retrieved from the memory store ($\mathcal{K}_s$) has poor relevance to the current search context, leading to spurious augmentation that distracts rather than helps the LLM.

To investigate this hypothesis further, we conduct an auxiliary evaluation that asks human annotators to rate the information retrieved from knowledge stores for a particular search context (see Section 3.2.2 for details). Specifically, we report Retrieval Relevance from both instantiations of our knowledge stores in Table 3. This metric is the average score from a Yes/No question - whether the retrieved context is relevant to the current search context (1) or not (0). As shown in Table 3, the quality of retrievals from the entity-centric knowledge store are significantly better than those from the linear search history-based store. This is because we have far greater control with entities being the atomic units of the knowledge representation space, and are able to *exactly* match entities in the context against entities in the store, rather than rely on a similarity-based retrieval process with the dense retriever [42].

### 5.1 Additional Studies and Analyses

**Ablation over Entity Matching Strategies.** Recall that our full K-LaMP relies on a combination of matching and retrieval of several different types of entities from its entity-centric knowledge store, namely: familiar, unfamiliar and lapsed entities (see Section 3.2.2). In order to individually measure the contribution of each strategy, we generate knowledge-augmented query suggestions on 313 search contexts using only one type of entity and ask human annotators to evaluate the results on Validity, Relatedness, and Usefulness. The

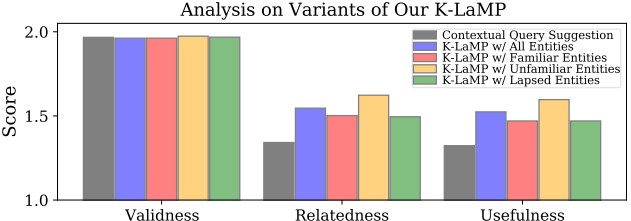

**Figure 4: Results of variants of our K-LaMP on personalized-knowledge retrieval strategy and results without retrieval.**

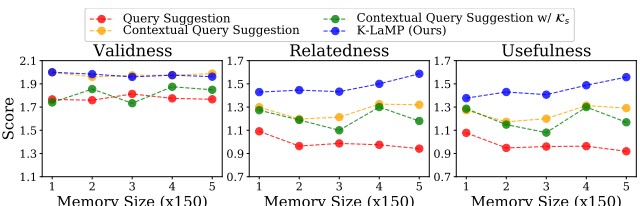

**Figure 5: Results of baselines and our K-LaMP with different numbers of previous queries (i.e., different memory sizes).**

comparative results are presented in Figure 4. Firstly, they reaffirm the fact that Validity is practically invariant to the choice of knowledge ingestion, since personal information does not affect whether a query is *valid* or not. The Relatedness and Usefulness metrics, however, are clearly impacted by the choice of entity matching strategy in consistent ways. In particular, using only "unfamiliar" entities yields the highest results across both metrics, even outperforming the full K-LaMP model. This seems to suggest that queries stemming from new (to the user) entities, which implicitly encourage exploration, are preferred over queries that revisit familiar ground.

While these results are true on this data, we note that real-world users may approach web search with different goals in mind: such as research, exploration, or revision, and it is difficult for judges to assess these goals from limited albeit rich data. As a result, we argue that having a comprehensive approach to capturing different views of the users' knowledge so that it may be deepened, expanded, or revived as the use-case may demand, is a robust strategy.

**Analysis over Interaction History Length.** A fundamental assumption in our LLM output personalization setup is that we can learn about users as they interact with search engines. A natural follow-up question to this assumption is to ask how the performance of systems that rely on personal knowledge change as a function of the length of the interaction history. To answer this question, we conduct an analysis with varying history lengths and report the results in Figure 5. From this, we observe again that Validity is not affected by the length of the interaction history, while Relatedness and Usefulness are. In particular, K-LaMP is the only model demonstrating consistent improvement with longer interaction histories, showcasing it's ability to grow richer representations of personal interests and knowledge over time. A reasonable explanation for this increment is thanks to the aggregation strategy that happens in K-LaMP's entity-centric knowledge store, which contrasts with the linearly stored histories of the other approaches.

**Analysis using different LLMs.** Finally, we conduct an auxiliary analysis to see how the quality of query recommendations

---

[3]We eschew more recent LM-based entity linkers [21, 41] since they have restrictive input token limits that are often exceeded by inputs in our scenario.

**Table 4: Results with different LLMs: GPT-3.5 and GPT-4.**

| Methods | LLMs | Validness | Relatedness | Usefulness |
|---|---|---|---|---|
| Query Suggestion | GPT-3.5 | 1.767 | 1.077 | 1.069 |
| | GPT-4 | 1.747 | 1.080 | 1.060 |
| Contextual Query Suggestion | GPT-3.5 | 1.967 | 1.177 | 1.202 |
| | GPT-4 | 1.987 | 1.367 | 1.313 |
| K-LaMP (Ours) | GPT-3.5 | 2.000 | 1.279 | 1.303 |
| | GPT-4 | 1.983 | **1.653** | **1.600** |

**Table 5: Results with automatic evaluation metrics.**

| Types | Validness | Relatedness | Usefulness |
|---|---|---|---|
| **Correlation w/ Human Evaluation** | **0.445** | **0.397** | -0.016 |
| Query Suggestion | 1.784 | 1.189 | 0.882 |
| Contextual Query Suggestion | 1.891 | 1.340 | 0.831 |
| Contextual Query Suggestion w/ $\mathcal{K}_s$ | 1.828 | 1.271 | 0.847 |
| K-LaMP (Ours) | 1.910 | 1.472 | 0.845 |

from different systems change if an LLM other than GPT-4 is used. Specifically, we use the July 01, 2023 version of GPT-3.5-Turbo as the LLM on 128 sets of query suggestions from two baselines (Query Suggestion and Contextual Query Suggestion) and our K-LaMP framework, and compare the results with GPT-4; these are shown in Table 4. Firstly, Query Suggestion is agnostic to the choice of LLMs, while Contextual Query Suggestion and K-LaMP are not. This is likely due to the fact that the latter two approaches must incorporate information from full web-pages as context and therefore benefit from the representational capacity of the larger model. More relevant to the contributions in this paper, we find that, even with GPT-3.5-Turbo, our K-LaMP approach shows comparable performance on the Usefulness metric with the second best model – Contextual Query Suggestion – despite the latter using GPT-4. This demonstrates the significant edge that an entity-centric representation of a user's personal interests and knowledge provides, for knowledge-augmented personalization of LLMs outputs.

## 5.2 Automatic Evaluation Setup

While human evaluation is useful for measuring systems and gaining insights, especially on a new task like the one we introduce, the process is slow and expensive, and therefore not scalable to bigger datasets, or future extensions. To address these issues, we explore an initial set of automatic evaluation metrics mirroring the ones described in Section 4.3 that may be used in the absence of human judgement. Recall that even human evaluation for tasks that deal with personalization is non-trivial; therefore, automatically evaluating the outputs of a contextual query system while conditioning on complex personal preference and knowledge data is very difficult.

Nevertheless, we propose and experiment with the following automatic formulations: (1) **Validity** – we compute the similarity between the query suggestion output of a system and the top search result (title and snippet) returned from issuing that query to the web search engine (to see if the query yields reasonable search results); (2) **Relatedness** – we measure the similarity between the query suggestion and the set of contextual personal entities retrieved from the user's entity-centric knowledge store (to ensure that the query is grounded in the personal context of the user); (3) **Usefulness** – we calculate the similarity between the query suggestion and the real subsequent queries that the user ended up issuing (in order to compare recommendations against the user's true actions). In each of these three metrics[4], similarity is computed by calculating the dot product of representations obtained from Contriever [42].

We validate these automatic evaluation metrics by ranking the systems on the test set, then computing Spearman's correlation

---

[4]We don't specify an automatic measure of Ranking, since this can be done trivially by scoring then sorting systems by one or more of the other automatic metrics.

against the ranking obtained by human judgement scores. As shown in Table 5, we find a moderate correlation on Validity and Relatedness, indicating that our proposed automatic metrics for these measures may be used as proxies in the absence of human labeling. However, there is no correlation between automatic and human Usefulness metrics. This is expected since (contextual) query recommendation is not expected to align perfectly with user behavior; users *should* be surprised and delighted by suggestions they would not have otherwise thought about.

There are several ways to improve the process of automatically evaluating contextual query suggestion. For example, we could use another LLM to perform a rubric-based evaluation of Validity, Relatedness and Usefulness, relying on it's capacity to account for complex personal and preferential data. Or we could train parametrized versions of the automatic metrics we have proposed on manually labeled data with the goal of increasing correlation with human judgement. We leave these and other explorations to future work.

## 6 CONCLUSION

In this work, we proposed a knowledge-augmentation framework for LLM output personalization called K-LaMP, that leverages historical user interactions with a search engine. The core of the personal knowledge we used for LLM augmentation relies on a novel lightweight entity-centric personal knowledge store that is constructed from the queries that users issue and the web-pages they viewed as they search and browse the web. To stress-test our personalization framework, we focused on the novel task of contextual search query suggestion, which crucially requires modeling both the contextual interests and the knowledge of users. Through human evaluation on an extensive test set, we showed that our entity-centric knowledge-augmented LLM produces personalized query recommendations that are better related to user's intent, more useful, and consistently ranked above those produced by several other LLM-powered query suggestion models. Our findings show that entities are effective atomic units for the representation of personal knowledge, offering a robust middle-ground of performance, flexibility, privacy and scalability, when compared with other personalization approaches that rely either on deep profile building or simple linearization of a user's historical interactions. We believe that K-LaMP has the potential to greatly impact both future research and product innovation. The use of personalized knowledge-augmentation for other search tasks such as snippet generation or question answering, the incorporation of other sources of data such as shopping or media-consumption histories, and the application to domains outside of search such as personal AI assistants, are all exciting avenues of future work. At the same time, enhanced evaluation remains an important future goal, with improved automatic metrics and real-world flighting as potential directions for exploration.

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

# A PROMPTS

In this section, we provide the prompts that we use for eliciting the responses from LLMs for personalized contextual query suggestion. In particular, we provide the prompt for our K-LaMP framework in Figure 6, as well as the prompts for other models, namely query suggestion, contextual query suggestion, and contextual query suggestion w/ $\mathcal{K}_s$ in Figure 7, Figure 8, and Figure 9, respectively.

---

*[System Message]*
You are an AI assistant whose primary goal is to suggest a next search query, in order to help a user search and find information better on the search engine. Two different queries and entities are separated by the token '|'. For example, 'Microsoft' and 'Google' would appear as 'Microsoft' | 'Google'.

---

*[User Message]*
You are going to suggest a search query that the user would search next based on the current query, the current session, the current article, and the personal entities.

The explanations of the query, session, article, and personal entities are as follows:
- The query is a specific set of phrases that the user enters into the search engine to find the information or resources related to a particular topic, question, or interest.
- The session refers to a sequence of queries requested by the user on the search engine, within a certain period of time or with regard to the completion of a task.
- The article refers to a specific webpage that the user clicks and reads from several search results displayed by the search engine in response to the requested query.
- The personal entity refers to a topic, keyword, person, event, or any subject that is specifically relevant or appealing to the individual user based on their personal interests.

Read the following query, session, article, and personal entities of the user as the context information, which might be helpful and relevant to suggest the next query.
Query: {Query}
Session: {Session}
Article Title: {Article['Title']}
Article Text: {Article['Text']}
Personal Entities: {Entities}

Based on the above query, session, article, and personal entities, please generate one next query suggestion with the rationale, in the format of

Query Suggestion:
Rationale:

**Figure 6: A prompt that we use in our K-LaMP model.**

---

*[System Message]*
You are an AI assistant whose primary goal is to suggest a next search query, in order to help a user search and find information better on the search engine. Two different queries and entities are separated by the token '|'. For example, 'Microsoft' and 'Google' would appear as 'Microsoft' | 'Google'.

---

*[User Message]*
You are going to suggest a search query that the user would search next based on the current query, and the current session.

The explanations of the query, and session are as follows:
- The query is a specific set of phrases that the user enters into the search engine to find the information or resources related to a particular topic, question, or interest.
- The session refers to a sequence of queries requested by the user on the search engine, within a certain period of time or with regard to the completion of a task.

Read the following query, and session of the user as the context information, which might be helpful and relevant to suggest the next query.
Query: {Query}
Session: {Session}

Based on the above query, and session, please generate one next query suggestion with the rationale, in the format of

Query Suggestion:
Rationale:

**Figure 7: A prompt that we use in the query suggestion baseline.**

---

[System Message]
You are an AI assistant whose primary goal is to suggest a next search query, in order to help a user search and find information better on the search engine. Two different queries and entities are separated by the token '|'. For example, 'Microsoft' and 'Google' would appear as 'Microsoft' | 'Google'.

---

[User Message]
You are going to suggest a search query that the user would search next based on the current query, the current session, and the current article.

The explanations of the query, session, and article are as follows:
• The query is a specific set of phrases that the user enters into the search engine to find the information or resources related to a particular topic, question, or interest.
• The session refers to a sequence of queries requested by the user on the search engine, within a certain period of time or with regard to the completion of a task.
• The article refers to a specific webpage that the user clicks and reads from several search results displayed by the search engine in response to the requested query.

Read the following query, session, and article of the user as the context information, which might be helpful and relevant to suggest the next query.
Query: {Query}
Session: {Session}
Article Title: {Article['Title']}
Article Text: {Article['Text']}

Based on the above query, session, and article, please generate one next query suggestion with the rationale, in the format of

Query Suggestion:
Rationale:

**Figure 8: A prompt that we use in the contextual query suggestion baseline.**

---

[System Message]
You are an AI assistant whose primary goal is to suggest a next search query, in order to help a user search and find information better on the search engine. Two different queries and entities are separated by the token '|'. For example, 'Microsoft' and 'Google' would appear as 'Microsoft' | 'Google'.

---

[User Message]
You are going to suggest a search query that the user would search next based on the current query, the current session, the current article, and the related article.

The explanations of the query, session, article, and related article are as follows:
• The query is a specific set of phrases that the user enters into the search engine to find the information or resources related to a particular topic, question, or interest.
• The session refers to a sequence of queries requested by the user on the search engine, within a certain period of time or with regard to the completion of a task.
• The article refers to a specific webpage that the user clicks and reads from several search results displayed by the search engine in response to the requested query.
• The related article refers to a specific webpage that the user had previously read with interest, which may be relevant to the current query, session, and article.

Read the following query, session, article, and related article of the user as the context information, which might be helpful and relevant to suggest the next query.
Query: {Query}
Session: {Session}
Article Title: {Article['Title']}
Article Text: {Article['Text']}
Related Article Title: {RelatedArticle['Title']}
Related Article Text: {RelatedArticle['Text']}

Based on the above query, session, article, and related article, please generate one next query suggestion with the rationale, in the format of

Query Suggestion:
Rationale:

**Figure 9: A prompt that we use in the contextual query suggestion w/ $\mathcal{K}_s$ model.**

