# OpenReview forum: "Knowledge-Augmented Large Language Models for Personalized Contextual Query Suggestion"
_ACM.org/TheWebConf/2024/Conference — TheWebConf24_

### Official Review · Reviewer_hv8z · 2023-11-14

**Novelty:** 4
**Technical Quality:** 4

**Review:**

The paper proposes a novel approach to personalize the outputs of Large Language Models by augmenting them with relevant context from users' interaction histories with a search engine.

The main contributions of the paper are the construction of an entity-centric knowledge store for each user and the use of this store to provide contextually relevant LLM prompt augmentations, leading to improved search experiences. Specifically, this paper constructs an entity-centric knowledge store for each user based on their search and browsing activities on the web, which is then leveraged to provide contextually relevant LLM prompt augmentations. This knowledge store is light-weight, since it only produces user-specific aggregate projections of interests and knowledge onto public knowledge graphs, and leverages existing search log infrastructure, thereby mitigating the privacy, compliance, and scalability concerns associated with building deep user profiles for personalization.

This paper also presents a thorough evaluation of the proposed approach using human judges and provides insights into the strengths and limitations of the method.

Reasons To Accept:

1. This paper presents a well-motivated and novel approach to personalizing Large Language Models that is grounded in user data and context.
2. The results demonstrate the effectiveness of their approach.
3. The paper is well-written and clearly organized, making it easy to follow the authors' arguments and contributions.
4. The authors construct new datasets, new metrics and human judges for better evaluation, which makes a big contribution to this field.

Reasons To Reject:

1. Lack of important description of methods. Entity extraction should be a core step of K-LaMP, however, it is too simple to say “by using a state-of-the-art entity linking system to tag and canonicalize the entities” in line 452. More details should be added.
2. Some concepts lack more detailed explanations, which can be confusing, such as “k-anonymization” in line 469.
3. Manual evaluation has a certain degree of uncertainty. More details should be supplemented to demonstrate the accuracy of manual evaluation.

**Questions:**

1. Does this paper consider the ambiguity of entities when building the entity-based knowledge store? For example, “Apple” may be a kind of fruit or a corporation, which affects the output of K-LaMP.
2. This paper appears to be only studying the results of single round recommendations. What would happen if there were multiple rounds of recommendations?
3. This paper claims that the method proposed in this paper is a lightweight method. Where does the method reflect that it is lightweight? Will this method become more complex as historical records grow?

**Ethics Review Description:**

There is no ethics issue

**Reviewer Confidence:**

3: The reviewer is confident but not certain that the evaluation is correct

**Scope:**

4: The work is relevant to the Web and to the track, and is of broad interest to the community

---

### Official Review · Reviewer_DDtL · 2023-11-24

**Novelty:** 6
**Technical Quality:** 4

**Review:**

This paper proposes the K-LaMP, a knowledge-augmentation framework for Language Model (LLM) output personalization. It leverages historical user interactions with a search engine to create a lightweight entity-centric personal knowledge store. The focus is on contextual search query suggestions, where the goal is to provide personalized query recommendations based on user context and knowledge. Human evaluation shows that K-LaMP consistently outperforms other LLM-powered query suggestion models in terms of relevance, usefulness, and ranking.

This study aims to validate the approach within the context of query suggestion and presents promising results. The paper is well-structured, providing detailed methods and illustrative figures. It conducts extensive experiments that demonstrate the method's effectiveness. However, the authors did not provide a sufficiently detailed explanation of the data source and processing. As a result, it is challenging to assess the accuracy of the metrics chosen by the authors and the results of human evaluation.

**Questions:**

1. As a user's query history grows and potentially exceeds the input token limit of the LLM, how does K-LaMP handle this situation?

2. The biggest concern in this research is that in real-world scenarios, a user's search interests evolve over time. Relying solely on previous personalized search records can lead to the problem of creating filter bubbles and potentially causing user fatigue. Can K-LaMP introduce mechanisms for exploration to extend the current framework?

**Reviewer Confidence:**

2: The reviewer is willing to defend the evaluation, but it is likely that the reviewer did not understand parts of the paper

**Scope:**

3: The work is somewhat relevant to the Web and to the track, and is of narrow interest to a sub-community

---

### Official Review · Reviewer_VdSE · 2023-11-27

**Novelty:** 5
**Technical Quality:** 5

**Review:**

The paper introduces a novel approach to personalize Large Language Models (LLMs) for web search by leveraging users' interaction histories. It proposes constructing lightweight, user-specific entity-centric knowledge stores based on search and browsing activities. These knowledge stores are used to augment LLMs with contextually relevant prompts, improving search experiences by tailoring outputs to users' preferences and historical knowledge. Experimental results demonstrate the effectiveness of the approach, outperforming other LLM-powered baselines in generating contextually relevant and personalized query suggestions.

strengths：
1.	K-LaMP is an effective lightweight method that enhances the performance of personalized contextual query generation by incorporating entity knowledge from search history.

weaknesses:
1.	One notable drawback of the method is its assumption that all entities appearing in the search logs are of interest to the user. This assumption may be considered unreasonable, as not every entity encountered in search history necessarily reflects the user's genuine interests.
2.	A notable weakness in the experimental design pertains to the baseline configuration. Specifically, the retrieval step of K_s relying solely on the user's current query is deemed both unreasonable and unfair. This limitation is elucidated by the observed decline in results when utilizing K_s, as demonstrated in Table 1. The inadequacy of this baseline configuration compromises the validity of the comparison, as it introduces an inherent bias by neglecting the contextual richness provided by the user's interaction history and context.

**Questions:**

1.	As previously highlighted, not all content browsed by users necessarily aligns with their specific interests. Could there be a consideration for employing a more granular memory approach when utilizing K_s?
2.	During the evaluation, the assessors were provided with the 30 most frequent entities from the user's personal entity-centric knowledge store. This practice raises concerns about the fairness and effectiveness of the evaluation process. Are there alternative methods for evaluation, such as pre-labeling user interests or ensuring consistency between the assessors involved in evaluation and annotation?

**Reviewer Confidence:**

3: The reviewer is confident but not certain that the evaluation is correct

**Scope:**

4: The work is relevant to the Web and to the track, and is of broad interest to the community

---

### Official Review · Reviewer_3bYp · 2023-11-29

**Novelty:** 3
**Technical Quality:** 4

**Review:**

This paper adopts LLMs for personalized query suggestion, where the personalization is based on previous search queries as well as browsed web pages. The method extracts entities from queries/webpages and uses embeddings to find the ones that are more relevant to the current query in-session for personalization.

Strength:
- Conducted both human eval as well as auto eval for validating the results.
- In general well-written and easy to follow.

Weakness:
- Technical contributions might be limited (and not meeting the bar for WWW). The key idea in this paper is to use embeddings for retrieving relevant user information from search and browsing history, which is pretty common practice in my opinion.

- Baselines the paper focuses on prompting LLMs for query suggestions, but did not compare against other baselines.

**Questions:**

n/a

**Reviewer Confidence:**

2: The reviewer is willing to defend the evaluation, but it is likely that the reviewer did not understand parts of the paper

**Scope:**

4: The work is relevant to the Web and to the track, and is of broad interest to the community

---

### Decision · Program_Chairs · 2024-01-22

**Decision:**

Accept

**Comment:**

This work proposes to augment the input of large language models (LLMs) with entity knowledge from user behavior history. The web application is contextual query suggestion. The method was clearly presented, and the experimental results were capable of demonstrating the usefulness of knowledge augmentation. Reviewers identified the technical quality and novelty though held relatively low confidence. The reason is that the dataset is anonymized, the topic is new, and very few organizations are able to perform such a study, considering the requirement of data and computational resources. This work successfully demonstrated the effectiveness of knowledge augmentation on using LLMs for contextual query suggestion. Considering the prior work, it would be great if the study could incorporate a comparison against existing work (non-LLMs) to address the importance of "context" in query suggestion. If context is truly helpful, LLM is truly powerful, and the proposed method is truly useful, the community will appreciate the contributions of this work.